# Perceived Stress Is Directly Associated with Major Consumption of Sugar-Sweetened Beverages among Public University Students

**DOI:** 10.3390/bs13030232

**Published:** 2023-03-07

**Authors:** Cesar Campos-Ramírez, Jorge Palacios-Delgado, Maria del Carmen Caamaño-Perez, Nicolas Camacho-Calderon, María Elena Villagrán-Herrera, Adriana Aguilar-Galarza, Teresa García-Gasca, Miriam Aracely Anaya-Loyola

**Affiliations:** 1Department of Neurometabolism Sciences, School of Medicine, Autonomous University of Queretaro, Clavel 200, Prados de la Capilla, Queretaro 76176, Mexico; 2Neuroeconomics Research Department, University of Valle de México, Campus Querétaro, Blvd. Juriquilla 1000 A. Santa Rosa Jáuregui, Querétaro 76230, Mexico; 3Department of Human Nutrition, School of Natural Sciences, Autonomous University of Queretaro, Av. De las Ciencias Juriquilla, Queretaro 76230, Mexico

**Keywords:** perceived stress, sugar sweetened beverages, university students, consumption

## Abstract

Stress is a condition that has been related to the development of risk behaviors for health such as sugar-sweetened beverages (SSBs) consumption. The aim of this study was to examine the link between SSBs consumption and perceived stress level in university students. This was an observational, cross-sectional and single-time-point study where the subjects were recruited as a non-probabilistic sample of first-year university students. The students reported their SSBs consumption through a validated questionnaire, as well as their perceived stress level, evaluated through the Cohen scale. Comparisons were made between the means of all variables. Factorial analysis of variance was conducted to explore the effect of the variables’ interaction on the stress level. One-way analysis of variance was performed to assess differences between the sexes. Men consumed more SSBs (6101.17 ± 3772.50 mL/week) compared to women (4294.06 ± 3093.8 mL/week). However, women had higher scores of perceived stress and showed a strong association of stress with the SSBs consumption pattern (r and *p*-value). This study shows for the first time the association that exists between stress and SSBs consumption and indicates that it is related to sex in the young population.

## 1. Introduction

Obesity is an increasingly major health issue in developing countries. This condition can be driven by a complex set of factors including inherited (genetic) and adaptive (epigenetic) biological traits, behavioral and emotional life experiences, as well as environmental and social variables. Nevertheless, the impressive increase in obesity rates over the past few decades, in which we dramatically changed our natural and food-related environment, strongly suggests that obesity is mainly driven by environmental adaptive, rather than genetic inherited factors [1,2].

On the other hand, energy intake has also increased in our modern society through the ready availability and low cost of energy-dense foods and drinks [3], such as nutrient-poor foods and, especially, sugar-sweetened beverages (SSBs), which are highly palatable products (HPP) that can be experienced with intense pleasure. Interestingly, HPP consumption is upregulated by the intensity and duration of the stress experience of the organism [4,5,6]. Also important is to note that the pleasure experience associated with palatability is considered necessary and sufficient for stress dampening [7]. In other words, we eat and drink these products to cope with our stress.

Stress is a complex set of the organism’s responses to stimuli or events that are interpreted as factors jeopardizing its psychological and/or physical integrity. This complex response includes biological, cognitive and behavioral actions, involving the autonomic, neuroendocrine, metabolic and immune systems [8]. When this response allows achieving an integral balance, the organism is able to learn, and this adaptation makes it more able to confront with the same or a similar treat with ease and even to confront harder challenges [9].

During this response, the organism typically develops vulnerabilities expressed in the progression of several types of psychiatric conditions, such as addiction, anxiety and depression, but also in the development of chronic diseases [10,11].

More important, metabolic stress has also been linked to situations in which food is restricted or consumed in excess [7]. Similarly, caloric restriction and/or fasting, as metabolic stressors, promote a resilient metabolic adaptation, while an excess of food or drink consumption promotes maladaptation and the development of vulnerabilities [12].

All these adaptations occur mainly throughout the individual history of food and drink consumption, and these types of biological adaptations are currently studied by two modern disciplines, i.e., nutrigenetics, which studies how genetic variations affect the organism response to a given diet, and nutrigenomics, which focuses on how dietary components affect the genome function. For example, it has been demonstrated that dietary components of popular beverages, such as green tea, red wine or SSBs, affect different metabolic pathways [13]. SSBs consumption interacts with genetic signals associated with obesity risk factors; therefore, a genetic predisposition towards a higher body mass index (BMI) in people showing a higher consumption of SSBs was found [14]. Thus, this association widens the differences between individuals with and without genetic predisposition to obesity [15]. However, the consumption of food and drinks also modifies gene expression through epigenetic changes [16].

Moreover, mounting evidence indicates that a frequent SSBs consumption promotes metabolic and inflammatory abnormalities [17]. Environmental changes that affect genome regulation, also called epigenetic changes, are commonly associated with diet and can be considered as an example of cellular plasticity of different tissue and organs, affecting their function and occurring throughout the individual diet history. Among the many organs, these adaptations affect the central nervous system, consequently regulating food and drink intake as well as appetite/satiety regulatory mechanisms, but this goes along with metabolic adaptations, such as lipid accumulation, as well as immune, endocrine and autonomic functions [18].

According to the scientific literature, stress and SSBs consumption are closely associated. SSBs can act as stress regulators directly on the brain reward system, which is involved in the eating behavior and influences metabolic or biological stress responses [4,5,6], as well as the effect of SSBs consumption in stress dampening [7].

In this study, we were interested in understanding the link between SSBs consumption and the stress response in Mexican freshmen university students, which are part of one of the populations with the highest SSBs consumption worldwide [13]. Particularly, we examined the consumption per week of the different SSBs groups available in the market, as well as if their association with the stress response experience differed by sex. It is of vital importance to highlight that there are no studies that address the relationship between these two factors in the Latin population from Mexico, which is of great interest since it is the population with the highest consumption of SBBs worldwide. In addition, the present investigation has an added value because the findings could be extrapolated to other populations worldwide, since the effects of SSBs consumption on human physiology do not differ in a general way between races.

## 2. Materials and Methods

### 2.1. Participants

This was an observational, cross-sectional and single-time-point study where the subjects were recruited by non-probability sampling of first-year students from the Autonomous University of Queretaro (UAQ) located in central Mexico during the fall of 2019. The recruitment of students was through the “Su Salud” program, which is a comprehensive health program that performs an integral assessment of all freshmen students of UAQ. The inclusion criteria were: (a) to be an enrolled student in one of the schools in UAQ participating in the Su Salud program; (b) to have signed the informed consent letter; (c) to attend the instructional talk for the study and (d) to answer all the questionnaires. The exclusion criteria were: (a) the omission of or a partial questionnaire response and (b) the pre-existence of a medical condition disabling the evaluation of anthropometric (weight, total and visceral fat) or other measures. The study was approved by the ethics committee of the Natural Sciences Department, Autonomous University of Queretaro, with registration number 98FCN2017.

The freshmen that met the eligibility criteria and agreed to participate in the study were asked to sign the informed consent letter. The enrolled students did not obtain any type of remuneration, either monetary or in any kind, for their participation in the study. A total of 632 students participated in this study, 265 were men (42.0%), and 367 were women (57.9%).

### 2.2. Measures

The SSBs Consumption Questionnaire is a questionnaire based upon several other validated questionnaires [19,20,21,22]. It consists of a list of beverages grouped as follows: (1) cola soft drinks (caloric and non-caloric), (2) all other flavored soft drinks (caloric and non-caloric), (3) industrialized juices (caloric and non-caloric), (4) industrialized teas or infusions (caloric and non-caloric), (5) milk (whole, partially skimmed and skim), (6) sports drinks (caloric and non-caloric), (7) energy drinks (caloric and non-caloric) and (8) homemade sweetened drinks (caloric and non-caloric).

The SSBs consumption questionnaire asked each participant whether they consumed any of the different beverage groups in either their caloric or non-caloric versions. The SSBs consumption questionnaire gives the choice to select the portion (250 mL, 355 mL, 600 mL or 1 L) or commercial presentation based on the observed available drinking containers of the different products found in the university campus and in the stores nearby. This questionnaire was also designed to report SBBs consumption during an average week throughout the last month. To quantify the consumed amount of beverages, including non-caloric and caloric SSBs, the questionnaire responses were transformed into mL/week for each of the beverages groups (caloric and non-caloric) included in the questionnaire. Total liquids consumption was considered as the sum of the caloric and non-caloric versions of all beverages groups evaluated plus natural juices and non-sugar-sweetened beverage (NSSBs). NSSBs included natural water and infusions without any sweetener.

Stress scale. The Perceived Stress Scale (PSS) [23] was used to measure the perception of stress, which is the degree of appraised stress in situations of life. The items are developed to determine how unpredictable, uncontrollable and overloaded the respondents find their life’s events. The scale also includes direct queries about the current levels of experienced stress. The items are easy to understand, and the response options are simple to select. Importantly, the questions are of a general nature and, hence, are relatively free of content specific to any subpopulation group. The questions in the PSS ask about feelings and thoughts during the last month. In each case, the respondents were asked about how often they felt in a certain way.

The PSS total score is obtained by reversing the response (e.g., 0 = 4, 1 = 3, 2 = 2, 3 = 1 and 4 = 0) of the seven positively stated items (4, 5, 6, 7, 9, 10 and 13) and then summing these scores with those of the seven negative stated items (1, 2, 3, 8, 11, 12 and 14). The score obtained from the positive stated items represents the ability to control stressful or threatening situations, while that from the negative stated items represents the perception of helplessness or loss of control and its consequences. All items are valued on a five-point Likert-type scale (0 = never, 1 = almost never, 2 = occasionally, 3 = often, 4 = very often). The total scale has a range of scores from 0 to 56. It is important to note that the higher scores on the total PSS indicate higher levels of perceived stress. In order to assess the stress level in this sample, the 14-item PSS version was used, which was translated and adapted for the young Mexican population [24]. This adaptation was made after conducting a pilot study among university students, 20.48 years old (SD = 3.62), which showed adequate internal consistency (α = 0.83) and convergent validity with the Beck depression inventory (Beck, Ward, Mendelson, Mock, 1961)—rs = 0.553, *p* = 0.001—and with the emotional exhaustion scale (Ramos, Manga and Moran 2005)—rs = 0.521, *p* = 0.001—in addition to a confirmatory factor analysis with acceptable adjustment statistics (KMO = 0.87), which indicated that the cultural adaptation of the scale used in this study was satisfactory. Finally, the participants were classified according to the reported score into three categories: Low stress (0–18), moderate stress (19–36) and high stress (37–56).

Since the PSS scale scores and the SSBs consumption pattern appeared to be different in men and women, it was decided to carry out separate analyses. Based on the PSS total score, the participants were categorized in three different groups according to their stress level (low, moderate and high) [25]; then, comparisons were made for the consumption of each SSBs group with the category of stress as an independent variable.

Pearson correlations, odds ratio and quartile categorization were performed to assess the association between SSBs consumption and the PSS scores. The quartiles for women were established at Q1 ≤ 2000, Q2, 2001–3715, Q3; 3716–6010 and Q4 > 6010 mL/week, for men, the quartiles were established at Q1 ≤ 3340, Q2, 3341–5177, Q3, 5178–8395 and Q4 > 8395 mL/week.

Anthropometric data. All measures were performed by a trained nutritionist. Weight determination and body composition data were obtained using a multifrequency bioelectric impedance device (Seca mBCA 515, model 0123; Hamburg Germany). We used for this study the total body fat percentage (TBF%) and visceral fat percentage (VF%).

### 2.3. Statistical Analyses

Descriptive statistics for the PSS scores, as well as for the SSBs intake data, were performed. Data are presented in tables as the mean ± standard deviation or as percentages. Comparisons were made between the means of all the variables analyzed according to sex. Factorial analysis of variance was conducted to evaluate the effect of SSBs intake on the stress level and anthropometric variables. One-way analysis of variance was performed to assess differences between sexes. All analyses were completed in SPSS v22.

## 3. Results

### 3.1. Participants’ Charateristics

A total of 632 freshmen students participated in this study; 266 of them were men (42.0%), and 366 were women (57.9%). The participants came from different schools: Chemistry, Engineering, Languages and Letters, Natural Sciences and Political Sciences (Table 1). The participants had an age between 17 and 25 years, with a mean of 18.96 ± 1.52 years.

### 3.2. PSS Scores of Men and Woman

The factorial analysis of variance did not show a significant interaction between the effects of age, stress level and sex on SSBs consumption. The PSS scores were clearly different between sexes; the women’s scores revealed higher perceived stress according to the negative factors and the total score, but a lower score for the positive factors, (Table 2).

### 3.3. SSBs Scores of Men and Woman

Men reported a higher consumption of almost all SSBs than women, except for industrialized teas or infusions and drinkable yogurt, whose reported consumption was similar between men and women (Table 3).

### 3.4. Associations between SSBs Consumption and Perceived Stress

#### 3.4.1. Comparisons of SSBs Groups Consumption According to the Stress Level

For women, a significant difference was observed regarding industrialized juice and total SSBs consumption, with women who reported a higher stress level consuming statistically more of these beverages than women in the low stress level group. For men no differences in SSBs consumption were found among the stress level groups (Table 4 and Table 5).

#### 3.4.2. Comparison Using SSBs Quartile Categorization

The relation between stress level and SSBs total consumption quartiles was analyzed according to the sex of the participants (Figure 1).

A significant difference was observed for Q4 in the negative factors’ score, with 2.12 points more than for Q1, in women. For total stress, Q4 presented an average of 3.5 points more than Q1. No significant differences were found for men.

#### 3.4.3. Correlations

For women, significant correlations between SSBs total consumption and the positive factors’ score, the negative factors’ score and the total scale score were detected (Pearson R = −0.115 (*p* < 0.05); 0.137 (*p* < 0.01) and 0.156 (*p* < 0.01), respectively). No significant correlations were found between perceived stress and SSBs total consumption in men.

#### 3.4.4. Odds Ratio

Finally, an association was made between the SSBs consumption quartiles and the stress level reported by the students. Regarding the men, 31.5% were located in the group with high perceived stress, and the odds ratios for presenting beverage consumption within the highest quartile were 1.33 (95% CI: 0.742–2.395) compared to men who did not have high perceived stress. For women, 29.2% reported high perceived stress, and the odds ratios for presenting beverage consumption within the highest quartile were 2.007 (95% CI: 1.219–3.304) compared to women who did not have high perceived stress.

### 3.5. Stress and NCSBs Consumption in Men and Woman

Regarding the consumption of the NCSBs (artificial sweetened), we found that it was significantly lower than the consumption of SSBs. NCSBs consumption in men was 356.25 ± 942.12 (mean ±SD) mL/week, while the average in women was 440.41 ± 1065.66 mL/week. No associations were found between the level of stress and NCSBs consumption overall and within the two sexes.

### 3.6. Natural Juices, Total Liquids Consumption and Stress Association

#### 3.6.1. Natural Juices

The consumption of natural juices was on average in men of 559.00 ± 902.50 mL/week, while in women it was of 485.89 ± 802.25 mL/week. No association between stress and natural juice consumption was found in the overall participants.

#### 3.6.2. Total Liquids

The average of total liquids consumed by men was 12,564.18 ± 7215.45 mL/week. The SBBs consumption was 55.71% of the total liquids consumption for men and 48.86% for women. Additionally, for men, NCSBs overall mean consumption amounted to 5540.09 ± 5889.73 mL/week.

## 4. Discussion

Emotional well-being has multiple associated factors; however, it has been observed that one of the main factors in young population is stress, which is associated with (HPP) consumption and unhealthy eating habits [26]. The results of the present study showed a difference in the levels of perceived stress between sexes, with higher scores for women, and very similar levels to those found in other populations in the same age range [27].

When analyzing the perceived stress by categories, it was found that women tended to increase the consumption of most SSBs groups, but this increase was only significant for industrialized juices and total SSBs consumption, with, on average, 1.2 L higher consumption by women who presented a high perceived stress score compared to those with low perceived stress. These findings are consistent with previous reports where a positive correlation was found between stress values and sugar and fast-food consumption [27,28,29]. This can be explained by the fact that exposure to a stressor can cause impaired control in the mechanisms responsible for limiting the intake of high-sugar and high-fat foods, mainly in women [30]. Another important aspect to note is that the individuals who are susceptible to increasing their calorie and HPP intake in response to stress are those that normally maintain restriction diets [30]. In a study on a young population [27], correlations were found between high-sugar food consumption and the PSS score; however, when analyzing the association between high-sugar food consumption and depression symptoms, they found no association, which could be indicating that the high-sugar or HPP consumption could decrease the signs and symptoms of depression. However, the possible mechanisms involved is not clear, nor if SSBs consumption has the ability to prevent this mental illness.

To corroborate the association observed between perceived stress level and SSBs consumption, it was decided to group the participants in quartiles according to consumption. It is important to note that different values were designated in the construction of the quartiles for men and women, because the SSBs consumption patterns clearly differed between sexes, as in men, higher consumption was observed for the majority of the SSBs groups evaluated, as well as a consumption of almost 2 L higher when evaluating all SSBs consumed, which agrees with previous reports with similar populations [31].

The items belonging to the positive factors of the Cohen scale used in this work were focused on evaluating the individual control over stressful or threatening situations. Very similar constructs were previously evaluated. For instance, some studies [32,33] found the construct called “perceived behavioral control” (perception of the ease or difficulty to perform a behavior) was positively associated with soft drinks consumption in adolescents, while another study [34] found that a very similar construct, previously associated with healthy eating behaviors, i.e., self-efficacy (individual capacity to successfully execute a behavior), was the main predictor of soft drinks consumption in university students. Women belonging to the highest SSBs consumption quartile presented higher scale scores for negative factors and total scale, which corroborated a consumption trend depending on the levels of perceived stress. For men, none of the comparisons were significant. One of the possible explanations for this association is that stress-induced feeding has the ability to activate the reward system and decrease the hypothalamic–pituitary–axis (HPA) activity by releasing opioids [35]; in addition, HPP consumption activates a negative feedback mechanism from the HPA axis, mimicking the effect of stress recovery [36]. However, repeated exposure to this type of foods can lead to an excess of calorie intake and to obesity development [37]. The mechanisms that explain the connection/relation between stress and eating behavior are not fully characterized. Hormones such as leptin, insulin and glucocorticoids, which are involved in food intake regulation and energy balance in the hypothalamus, have been studied, and resulted to be important biomarkers of the mechanisms of this relation [38].

Further, it has been observed through functional magnetic resonance imaging that chronic stress in women causes brain activity increase in areas related to reward, motivation and decision making in response to visual stimuli of HPP; in addition, the same stimuli caused an activity decrease in strategic planning and emotional control areas, suggesting that chronic stress can alter the brain response to food in a way that predisposes individuals to unhealthy eating habits and obesity development [39]. The processing of emotions in areas such as the amygdala, striatum and hippocampus has also been significantly related to behavior-determined stress and reward [40], and dysfunctions in these structures could be related to eating disorders as well as to obesity and addictions [38]. Therefore, and in accordance with the results of this study, other factors such as emotional, motivational, and executive information from limbic system areas, the striatum, and the pre-frontal cortex may influence food intake, particularly, SSBs consumption. 

Although a relationship between stress and eating behavior in humans has been widely observed, the mechanisms by which these phenomenon are linked have not been fully understood. However, is accepted that some general characteristics such as being a woman, overweight or obese and having followed or following a restrictive diet are predisposing factors to overeating under stress conditions, that is, these characteristics give the individual greater reactivity to stress [41]. However, analyses in individuals on the calorie restriction regimens produced contradictory results [42]. Although this relationship may be bidirectional, the majority of individuals (70%) increase their caloric intake during or after stressful events [43]. Furthermore, a “medication” with HPP could have some beneficial effects on an individual functioning [44]. However, in the long term, it is a counterproductive strategy to relieve stress, since the high caloric content of SSBs contributes to obesity and therefore to the dysfunction of appetite-regulating hormones [45], inflammation [46] and various metabolic complications [47].

One of the main findings of this study is the difference in the manifestations of SSBs consumption between sexes. In men, it seems to be mainly associated with an increase in visceral fat, but not in women [48]. One explanation is related to the anthropometric differences between men and women, with men having less total fat and greater abdominal fat, and premenopausal women having greater subcutaneous femoral/gluteal fat [49]. When men present a higher SSBs consumption and therefore a higher sugar consumption, a positive association with visceral fat is expected, i.e., SSBs consumption would cause an increase in visceral fat in men and not in women. These differences in fat distribution are also attributable to hormonal differences, particularly of estrogens, which are found in much lower concentration in men [50]. Visceral adipose tissue has estrogen receptors whose activation mediate lipolysis via the activation of hormone-sensitive lipase in women; on the contrary, in men, this activation would have an antilipolytic effect, since this tissue does not have estrogen receptors. Therefore, the accumulation of visceral fat would be determined by the activation of lipoprotein lipase, which is the key enzyme for the accumulation of fat in this area [51]. The net result of the above is a lower accumulation of abdominal fat in women in response to an increase in caloric or sugar intake. Other factors such as greater release of leptin in the gluteal/femoral zone and the effects at the hypothalamic level on the regulation of appetite by estrogens would function as a protective for women who have less capacity to consume food in large quantities [52] and explains why a higher SSBs consumption in men is reflected in an increase in abdominal fat. Therefore, premenopausal women in general have a lower prevalence of metabolic diseases, since the visceral adipose tissue is associated with a higher associated of metabolic disorders than subcutaneous fat [49] due to the protective effect of ovarian hormones, which agrees with the results of this study.

Finally, when obtaining the odds ratios as an association measure for the studied factors, the theory of stress influencing SSBs consumption was reinforced, as we observed that women who self-reported a high perceived stress were more likely to belong to the highest SSBs consumption group.

## 5. Conclusions

This study showed for the first time the association that exists between stress and SSBs consumption in university students. This population is in an important period of biological and social maturation. Freshmen university students face important lifestyle changes due to their integration into university life, a time window with a great sensitivity to stressors and to the development of mental illnesses or risk behaviors that can last for the rest of their adult life [50,51,52]. This work shows an important area for the identification of risk factors and the subsequent implementation of prevention strategies for this phenomenon. Future global public policies should focus on the identification of mental disorders that affect the young population inside and outside universities, since in recent years this type of illness has been on a dramatic increase. In particular, it is proposed to use questionnaires similar to those used in this study for the early detection of these conditions in larger populations and thus establish strategies that truly have an impact on public health. It is important to highlight that the results presented here agree with reports from other countries besides Mexico; therefore, it is certain that the main recommendation is the drastic reduction in the consumption of any SSBs due to their wide and consistent side effects. The above can be used in clinical practice by health professionals and for the theoretical support of future research in the field.

The study design and the lack of measurements of other factors that influence SSBs consumption make the results unable to demonstrate causality. More research is needed on stress-mediated eating in order to fully understand its impact on risk behaviors development and diseases related to obesity.

## Figures and Tables

**Figure 1 behavsci-13-00232-f001:**
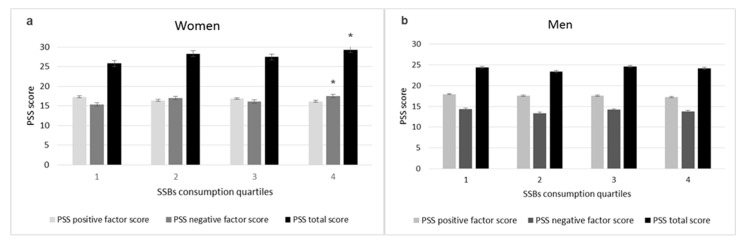
PSS score based on SSBs total consumption quartiles. The PSS score is reported as the mean ± standard error of each group differentiated by the positive factors’ score, negative factors’ score and total score. (**a**) The quartiles for women were established at Q1 ≤ 2000, Q2, 2001–3715, Q3, 3716–6010, and Q4 > 6010 mL/week. (**b**) For men, the quartiles were established at Q1 ≤ 3340, Q2, 3341–5177, Q3, 5178–8395, and Q4 > 8395 mL/week. * *p* < 0.05 with the Tukey test compared to the first quartile.

**Table 1 behavsci-13-00232-t001:** Sample distribution of the student participants according to their department of studies.

	Men (n = 266)	Women (n = 366)	Weighted (%)
Department			
Chemistry	55	63	18.7
Engineering	58	63	19.5
Languages and Letters	41	59	15.8
Natural Sciences	36	109	23.0
Political Sciences	76	72	23.0

**Table 2 behavsci-13-00232-t002:** Comparison by sex for the factors of the PSS scale.

	Men (n = 266)	Women (n = 366)	*p*-Value
	Mean ± SD	Mean ± SD	
Age	19.01 ± 1.52	18.92 ± 1.51	0.448
PSS positive factor score.	17.58 ± 4.60	16.74 ± 4.19	0.018
PSS negative factor score	14.05 ± 5.21	16.46 ± 4.84	0.001
PSS total score	24.26 ± 8.43	27.72 ± 7.89	0.001

PSS: Perceived Stress Scale. PSS total score: Higher score represents higher perceived stress. One- way analysis of variance.

**Table 3 behavsci-13-00232-t003:** Comparison of SSBs consumption in student participants by sex.

	Men (n = 266)	Women (n = 366)	*p*-Value ^1^
	Mean ± SD	Mean ± SD	
Cola soft drink	868.13 ± 980.94	510.10 ± 734.29	0.001
Other flavors soft drink	591.91 ± 757.57	292.03 ± 407.32	0.001
Industrialized juices	648.64 ± 836.23	463.34 ± 633.33	0.002
Industrialized teas or infusions	357.23 ± 595.49	340.16 ± 630.35	0.731
Milk	1408.04 ± 1513.81	1039.99 ± 1270.99	0.001
Drinkable yogurt	367.49 ± 541.81	342.24 ± 535.24	0.561
Sport drinks	368.08 ± 609.04	129.20 ± 360.13	0.001
Energy drinks	81.40 ± 264.07	22.21 ± 100.45	0.001
Home-made SSBs	1410.26 ± 1747.74	1154.80 ± 1380.00	0.041
SSBs total	6101.17 ± 3772.50	4294.06 ± 3093.84	0.001

SSBs: Sugar-Sweetened Beverages. ^1^ One-way analysis of variance.

**Table 4 behavsci-13-00232-t004:** Comparison of SSBs groups consumption in women according to stress level. n = 366.

		Mean ± SD	
	Low	Moderate	High
Cola soft drink	427.64 ± 642.42	479.63 ± 614.49	649.35 ± 934.84
Other flavors soft drinks	227.32 ± 343.27	308.21 ± 455.12	350.09 ± 406.98
Industrialized juice	339.60 ± 469.84	501.12 ±751.53	564.91 ± 621.09 *
Industrialized teas or infusions	353.00 ± 654.66	307.13 ± 600.21	369.72 ± 644.03
Milk	905.66 ± 1204.09	1004.25 ±1122.08	1185.98 ± 1386.52
Drinkable yogurt	323.40 ± 500.14	363.83 ± 575.99	340.42 ± 527.42
Sport drinks	167.00 ± 405.28	108.06 ± 337.02	112.71 ± 332.33
Energy drinks	23.36 ± 97.87	31.90 ±122.49	8.93 ± 67.11
Home-made SSBs	972.56 ± 1069.03	1110.82 ± 1339.44	1386.36 ± 1102.31
SSBs total	3739.54 ±2650.91	4214.95 ± 2879.34	4968.46 ± 3307.48 *

* *p* < 0.05 with the Tukey test compared to the low group.

**Table 5 behavsci-13-00232-t005:** Comparison of SSBs groups consumption in men according to stress level. n = 266.

		Mean ± SD	
	Low	Moderate	High
Cola soft drink	807.90 ± 809.97	907.29 ± 1000.17	877.08 ± 1120.02
Other flavors soft drink	573.86 ± 752.57	606.65 ± 817.85	587.26 ± 696.96
Industrialized juice	617.50 ± 813.17	621.22 ± 793.27	704.23 ± 908.89
Industrialized teas or infusions	286.53 ± 417.28	335.00 ± 610.60	451.90 ± 717.12
Milk	1508.92 ± 1703.54	1299.68 ± 1327.06	1427.87 ± 1500.34
Drinkable yogurt	358.42 ± 493.93	326.44 ± 538.49	418.57 ± 591.69
Sport drinks	361.14 ± 592.19	437.61 ± 689.44	293.15 ± 519.58
Energy drinks	80.17 ± 296.19	70.00 ± 210.02	94.46 ± 283.17
Home-made SSB	1157.50 ± 1262.33	1504.31 ± 2018.97	1558.99 ± 1840.67
SSB total	5751.93 ± 3542.69	6108.19 ± 4046.98	6413.13 ± 3702.88

## Data Availability

The data generated in this investigation will be available upon request.

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
