# Peer review of "Perceived Stress Is Directly Associated with Major Consumption of Sugar-Sweetened Beverages among Public University Students"

_behavsci, 2023, doi:10.3390/bs13030232_

Round 1

Reviewer 1 Report

Dear authors

I reviewed the manuscript entitled “Perceived Stress is Directly Associated with Major Consumption of Sugar Sweetened Beverages Among Public University Students”. This study is of great and important topic. In my opinion, it is greatly fall within the scope of journal. It should also be mentioned that the manuscript has addressed an issue that is my favorite subject area. Reading and evaluating different parts of the paper shows that the authors have made a lot of effort to carry out this research endeavor. Their efforts have resulted in important and ground-breaking conclusions that can certainly be used by different end-users including teachers, educators, and university decision-makers and managers. This makes their results and conclusions more reliable and rigor. Therefore, I recommend this paper for publication. However, there are some points that should be addressed by the respected authors before consideration of the manuscript for publication in Behavioral Sciences. My main comments are as follows:

1.       In the end of introduction, the main research questions (sub-objectives) should be mentioned.

2.       Please highlight the most important originalities of the research in the end introduction.

3.       This study has been done in Mexico. However, I recommend the respected authors to highlight the global value of this research in the end of introduction section. Can the results of this study be applied in other areas and/or countries?

4.       The methods completely support the results.

5.       Results section has been written and articulated very well. However, some more discussions should be done.

6.       In discussion section please try to put your results in an international scope and then provide the readers with some useful global level recommendations. Also, in discussion section the respected authors should try compare their results with the results of other researchers in Mexico and other parts of the world.

7.       In conclusion section, I recommend the respected authors to mention the main take-home message of the research in a short paragraph.

8.       Please highlight the main limitation of your study and try to draw some future pathways for the future researchers.

9.       In conclusion section try to highlight the main contribution of your paper to the theory and practice.

Author Response

We appreciate the time and dedication that you used to review the manuscript. The recommendations kindly made by you were made and are detailed below.

Reviewer 1

In the end of introduction, the main research questions (sub-objectives) should be mentioned.

Reply: We agree with your comment, the sub-objectives were added at the end of the introduction.

 Please highlight the most important originalities of the research in the end introduction.

Reply: The originalities of the study were added at the end of the introduction section.

This study has been done in Mexico. However, I recommend the respected authors to highlight the global value of this research in the end of introduction section. Can the results of this study be applied in other areas and/or countries?

Reply:  Were add the recommendation at the end of the introduction section.

The methods completely support the results.

Results section has been written and articulated very well. However, some more discussions should be done.

Reply:  A detailed review of the discussion and results sections was carried out.

In discussion section please try to put your results in an international scope and then provide the readers with some useful global level recommendations. Also, in discussion section the respected authors should try compare their results with the results of other researchers in Mexico and other parts of the world.

Reply:  Throughout the discussion section reference is made to different studies in various countries and it was added the general recommendations. Reference is made to other studies carried out in Mexico in lines 326-332 and 385-387.

In conclusion section, I recommend the respected authors to mention the main take-home message of the research in a short paragraph.

Reply:  Take-home messages were added in lines 427-430.

Please highlight the main limitation of your study and try to draw some future pathways for the future researchers.

Reply:  Lines 433-437 refer to the main limitations of the study and future lines of research.

In conclusion section try to highlight the main contribution of your paper to the theory and practice.

Reply:  We agree, the suggestion was added in the conclusions section.

Reviewer 2 Report

Line 55: Please change ‘has also increases’ to ‘has also increased’

Line 170-171: Please change to ‘In order to assess the stress level in this sample, the 14-item PSS version was used, which was translated and adapted for the young Mexican population.

Line 219: please amend to ‘effects of age…’

Line 303: please amend to ‘….however it has been observed…’

Author Response

We appreciate the time and dedication that you used to review the manuscript. The recommendations kindly made by you were made and are detailed below.

Reviewer 2

Line 55: Please change ‘has also increases’ to ‘has also increased’

Reply:  The change was made.

Line 170-171: Please change to ‘In order to assess the stress level in this sample, the 14-item PSS version was used, which was translated and adapted for the young Mexican population.

Reply:  We agree with the grammar correction, the change was made.

Line 219: please amend to ‘effects of age…’

Reply:  We agree with the grammar correction, the change was made.

Line 303: please amend to ‘….however it has been observed…’

Reply:  The change was made.

Reviewer 3 Report

the paper is improved but why two authors typed with red colors were added please refer to their urgent contribution, lines 64-70 need to be reorganized 

statistical analysis is somewhat difficult to understand please rewrite this paragraph in a correct manner

delete number 1 superscript for analysis of variance one way from tables and write it clearly in the section of statistical analysis

Author Response

We appreciate the time and dedication that you used to review the manuscript. The recommendations kindly made by you were made and are detailed below.

Reviewer 3

The paper is improved but why two authors typed with red colors were added please refer to their urgent contribution, lines 64-70 need to be reorganized .

Reply:  A detailed review of the authors contributions of this work was carried out, it was decided to change the order according to credit statement regulations. All authors agree with this change and the contribution can be seen in the authors contribution section.

Lines 64-70 need to be reorganized

Reply:  Changes were made to the structure of the paragraph, it was decided to keep the original idea to convey the generality of what the stress response currently involves.Statistical analysis is somewhat difficult to understand please rewrite this paragraph in a correct manner

Reply:  This section was rewritten for better understanding.

Delete number 1 superscript for analysis of variance one way from tables and write it clearly in the section of statistical analysis.

Reply:  It was decided to keep the number 1 superscript as the journal recommends it, however, the statistical analysis section was rewritten to clarify this section.

Reviewer 4 Report

Dear Editor,

Thank you for choosing me as a reviewer for the manuscript ID: behavsci-2156528 with title: Perceived stress is directly associated with major 2 consumption of sugar sweetened beverages among public 3 university students

Overview and general recommendation:

The main objective of the study was to examine the link 30 between SSBs consumption and the perceived stress level in university students. The article is well conceived. It contains all the necessary chapters. The introduction is written correctly and provides enough data to understand the goal of the research. The Material and methods are well explained. The methods are described correctly. The results are presented in Tables, are logical and can be followed with understanding. The results are discussed with the results of similar studies.

Specific comments:

-       Why was the age of the students not taken into consideration?

-       Why was the equal number of women and men not taken into consideration?

-       Most of the references in all the manuscripts are very old. New references must be added, especially in the last five years

-       The conclusions part is too long and should be shortened

-       The manuscript needs proofreading by native English speakers

-       The system for writing references must be standardized in accordance with the system of the journal

Author Response

We appreciate the time and dedication that you used to review the manuscript. The recommendations kindly made by you were made and are detailed below.

Reviewer 4

Why was the age of the students not taken into consideration?

Reply:  It was taken into consideration, however, there was no variation of any variable according to age, additionally the mean age remained constant between sexes.

Why was the equal number of women and men not taken into consideration?

Reply:  It was taken into consideration but given the study design it was not possible to maintain a proportional number of men and women, additionally the female population is higher in the university, see section Participant´s Charateristics

Most of the references in all the manuscripts are very old. New references must be added, especially in the last five years.

Reply:  Due it is a subject little studied in the country, there are very few current references, worldwide there are more studies related to stress that in fact are from the last five years and that were used as theoretical support, refs 6, 11, 18. Some of the knowledge of stress is a classical topic and for this reason references from 10-15 years ago were used, however we believe that the references used cannot be classified as “very old”.

The conclusions part is too long and should be shortened

Reply The conclusions section was modified, three of the reviewers recommended expanding it to include general recommendations

The manuscript needs proofreading by native English speakers

Reply:  Proofreading was carried out by a native English speaker in the last version of the manuscript.

The system for writing references must be standardized in accordance with the system of the journal

Reply:  The standardization of the references was carried out according to the norms of the journal

Round 2

Reviewer 1 Report

Dear authors

Thank you very much for addressing my concerns. I believe that the manuscript can be accepted for publication in present form. 

Regards,

Reviewer